# Assisted Reproductive Technology Results Using Donor or Partner Sperm: A Danish Nationwide Register-Based Cohort Study

**DOI:** 10.3390/jcm12072571

**Published:** 2023-03-29

**Authors:** Laura Catalini, Jens Fedder, Bente Mertz Nørgård, Line Riis Jølving

**Affiliations:** 1Centre of Andrology and Fertility Clinic, Odense University Hospital, 5000 Odense, Denmark; 2Research Unit of Gynecology and Obstetrics, Department of Clinical Research, University of Southern Denmark, 5000 Odense, Denmark; 3Center for Clinical Epidemiology, Odense University Hospital, 5000 Odense, Denmark; 4Research Unit of Clinical Epidemiology, Department of Clinical Research, University of Southern Denmark, 5000 Odense, Denmark

**Keywords:** live birth, donor sperm, partner sperm, intrauterine insemination, in vitro fertilization, fertility

## Abstract

This was a nationwide cohort study based on Danish health registers focusing on assisted reproductive technology (ART) treatments in women using donor or partner sperm from 2007 to 2017. Women using donor sperm were subdivided into groups based on relationship status: women with male partners, single women, or women with female partners. The live birth adjusted odds ratios (aORs) after the IUI treatments in women using donor sperm compared with women using partner sperm were 1.48 (95% CI: 1.38–1.59) in women with male partners using donor sperm, 1.20 (95% CI: 1.13–1.28) in single women, and 1.46 (95% CI: 1.32–1.62) in women with female partners. The live birth aORs after IVF treatments in women using donor sperm compared with women using partner sperm were 1.16 (95% CI: 1.02–1.32) in women with male partners using donor sperm, 0.88 (95% CI: 0.80–0.96) in single women, and 1.20 (95% CI: 1.00–1.44), in women with female partners. The use of donor sperm was associated with a higher chance of a live birth after the IUI treatments, but there was no difference after the IVF treatments. Our study invites healthcare professionals to increase their attention toward the different needs and fertility issues of all women attending fertility clinics.

## 1. Introduction

In the past, assisted reproductive technology (ART) was limited to heterosexual couples who were married or in stable relationships. Today, the number of single women, women with female partners, transgender men, and women receiving ART treatments is steadily increasing thanks to society’s legal and social changes and, with it, the demand for treatments with donor sperm [1,2,3]. Nevertheless, data regarding the efficacy of the current ART treatments with donor sperm in these patients are insufficient in the scientific literature. Most data are from intrauterine insemination (IUI) treatments, while almost no data are present for in vitro fertilization (IVF) treatments. The results are usually presented only based on partner or donor sperm without further differentiation [4]. Moreover, the few studies available do not consider potential confounding factors, such as age, smoking and drinking habits, previous fertility problems, or the number of prior ART treatments [5,6].

For single women and women in relationships with other women, at least the same ART success chances should be expected compared with women in relationships with men undergoing ART treatments. Indeed, there are no known biological differences due to sexual orientation, and they often attend fertility clinics to fulfill a pregnancy desire and not because they are infertile [5,6]. Some studies suggested that the fertility potential could be different in these population groups due to differences in the prevalence of obesity, sexually transmitted diseases, polycystic ovary syndrome, or smoking habits. However, further research is needed to confirm this [5,7].

The absence of the male gamete is considered the main missing factor in achieving a pregnancy for women without a male partner and those with a male partner who has to use donor sperm due to severe male infertility. For these patients, less invasive treatments, such as IUI with donor sperm, could be considered valid, even at older ages or for more attempts than for women using partner sperm. However, more data are needed to support this approach [8].

A factor not often considered is that using donor sperm could have some limitations. Indeed, in most ART treatments, only spermatozoa are utilized, while the rest of the seminal fluid is discarded. However, a growing number of studies suggested that previous exposure to male partner seminal plasma, the semen fluid component without sperm, could play an important role during IVF treatments in improving both implantation and pregnancy results [9,10,11,12]. Previous exposure to paternal seminal plasma influences maternal immunotolerance toward paternal antigens to protect the fetus from allorejection [13]. Moreover, regular contact with paternal seminal fluid seems to reduce the risk of gestational disorders, especially preeclampsia [9]. Indeed, a higher preeclampsia risk was observed in pregnancies obtained using donor sperm [14].

Our objective was to examine whether there is an association between the efficacy of ART treatments and the sperm used by comparing women using donor sperm and women using partner sperm. We subdivided the women receiving donor sperm according to the partner with whom they attended the clinic. Since they had different reasons for using donated sperm and could have different characteristics, such as age, weight, or infertility diagnosis, they could not be considered simply as one cohort.

We hypothesized that the better performance expected using donor sperm would not be confirmed, especially for IVF treatments. The cause could be the reduced or absent exposure to the paternal complete seminal fluid components involved in the implantation process, embryo immunorecognition, and pregnancy progression, which are probably more present in couples using their male partner’s sperm.

## 2. Materials and Methods

### 2.1. Study Design

Our study was a nationwide cohort study based on Danish health registries regarding ART treatments in all women using partner or donor sperm in Denmark from 2007 through to December 2017, including follow-ups on childbirths until the end of 2018.

In Denmark, all citizens have a unique civil registration number used in all Danish health registers for valid record linkage at an individual level [15]. The IVF register includes all ART treatments performed by public and private clinics. It was used to collect ART procedures; biochemical and clinical pregnancy data; and women’s characteristics, such as age, weight, infertility diagnosis, smoking, and drinking habits [16,17]. The Danish Medical Birth Register was used for data regarding the live birth results of ART treatments [18]. The Civil Registration System was used to collect data on death and immigration. It was necessary to exclude women who left the country or died before the end of the ART treatment to ensure complete follow-up data [15].

### 2.2. Setting and Study Population

In Denmark, a maximum of 6 IUI and a maximum of 3 IVF/intracytoplasmic sperm injection (ICSI) treatments with a fresh embryo transfer and an unlimited number of frozen–thawed embryo transfers (FETs) are offered for free by the national healthcare service if the woman’s age does not exceed 41 years in the public sector or 46 years in the private sector at the beginning of the treatment. From 2007, ART treatments were offered to all women without discrimination or additional evaluations related to marital status and sexual orientation [19]. According to the Danish guidelines and legislation, if donor sperm is used, recipients can have 6 IUI and 3 IVF treatments. If partner sperm is used, the recipients can have 3 IUI and 3 IVF/CISI treatments. Usually, IUI is offered as the first option unless there are ovarian or tubal problems or severely reduced semen quality. After the failure of the 3/6 IUI treatments, IVF/ICSI treatments are offered. In different-sex couples, treatments with donor sperm are offered to couples with azoospermia, which cannot be treated with testicular sperm extraction, or couples where the man is the carrier of a disease gene that the couple does not want to transmit to the offspring. In a few cases, treatments with donor sperm are also offered to couples with unsuccessful fertilization after IVF/ICSI.

The study population included all women registered in the IVF register with at least one IUI or IVF treatment cycle from 2007 to the end of 2017. We excluded egg donation cycles, cycles with preimplantation genetic testing, and egg recipient cycles. For this study, the definition of “IVF treatment cycle” included IVF and ICSI treatment cycles with a fresh embryo transfer or FET.

### 2.3. Exposed and Unexposed Cohorts

We divided the women in our study population according to the type of sperm used during the ART treatments:donor sperm for the exposed cohorts;partner sperm for the unexposed cohort.

Moreover, the population of the exposed women was further stratified according to the partner with whom they attended the fertility clinics, thus obtaining the following exposed cohorts:women with male partners using donor sperm;single women using donor sperm;women with female partners using donor sperm.

In the presence of inconsistencies between the type of sperm, partner sex, and primary diagnosis of infertility, or no information on the type of sperm or partner sex, we excluded that specific treatment cycle.

A flowchart showing the population selection process, including the reason for treatment exclusion, is presented in Appendix A.

### 2.4. Outcomes

The main outcome of interest was the number of live births per IUI or IVF treatment cycle counted as one event regardless of the number of babies born. Live birth was considered the result of a specific treatment cycle if the difference was 124–292 days from the day of IUI or IVF embryo transfer to birth. The range was selected to include all the possible viable births corresponding to 20–44 weeks of gestation [20]. We also evaluated biochemical and clinical pregnancy results per IUI or IVF treatment cycle as secondary outcomes, and their results are available in the Appendix A. Biochemical pregnancy was defined as a positive test for human gonadotropin (hCG) measured 14–16 days after IUI or IVF embryo transfer. Clinical pregnancy was defined as the presence of a gestational sac with a heartbeat, as evaluated using an ultrasound examination 7–8 weeks after the IUI or IVF embryo transfer.

The outcomes were calculated separately for IUI and IVF treatments because they are substantially different procedures.

### 2.5. Statistical Analysis

We constructed contingency tables for the main variables according to the different cohorts of exposed and unexposed women. Since each woman could have more than one ART treatment and possibly different partners, the unit of observation was an IUI or IVF treatment cycle.

The cohorts were compared using the non-parametric Wilcoxon rank-sum test or Kruskal–Wallis test for the continuous covariate age, and the chi-squared test was used for all the other binary and categorical covariates. A *p*-value < 0.05 was considered statistically significant.

Multilevel logistic regression analyses were performed to compute the crude and adjusted relative risk estimates as odds ratios (ORs) with 95% confidence intervals (CIs) for live births and biochemical and clinical pregnancies for each exposed cohort relative to the unexposed cohort. Furthermore, we accounted for possible multiple births or multiple IUI/IVF treatments by clustering for individual women.

All the analyses performed were complete case analyses. We had complete data regarding the treatment result for the crude OR, and we only considered treatments with complete data on all covariates for the adjusted OR.

### 2.6. Covariates

The following covariates were included in the regression models: woman’s age at the time of the IUI or IVF treatment cycle, calendar year of the ART treatment cycle, woman’s body mass index (BMI), woman’s smoking and alcohol intake at the time of the IUI or IVF treatment cycle, type of embryo transfer (fresh or frozen), and total number of treatments performed. We collected data on covariates from the IVF register.

There were missing data on BMI, smoking, and alcohol consumption. We performed sensitivity analyses, which involved including only one covariate at a time to assess the individual effect on the result. Age was the most influential variable (Appendix A).

The main regression models did not include data on female infertility diagnoses, such as endometriosis, polycystic ovarian syndrome, infectious diseases (HIV, hepatitis B or C), and poor sperm quality. For single women and women with female partners, the primary diagnosis was “without a male partner” most of the time, without any specification for other possible infertility problems. For this reason, a sensitivity analysis was performed, which included the infertility diagnosis as an additional covariate. The results are presented in the Appendix A.

Two other sensitivity analyses were performed to evaluate whether the semen quality of the male partner could impact the results. The variable “male infertility diagnosis” present in Appendix A was used to divide the male partners into two subgroups: partners with “normal sperm quality” and partners with “reduced sperm quality”. Reduced sperm quality included men with a diagnosis of aspermia, azoospermia, oliogozoospermia, oligoteratozoospermia, and retrograde ejaculation. Treatments where the male infertility diagnosis was unexplained, infection with HIV, missing, or previous infertility were excluded since there was no indication about sperm quality.

In the first analysis, the unexposed cohort only included women with male partners with normal sperm quality. In the second analysis, the unexposed cohort only included women with male partners with reduced sperm quality. The results are presented in the Appendix A.

## 3. Results

### 3.1. Patient Characteristics

The study included all ART treatments in women using partner (IUI, n = 80,949; IVF, n = 74,425) or donor sperm from 2007 until the end of 2017. ART treatments in women using donor sperm were subdivided into different-sex couples (IUI, n = 8780; IVF, n = 1937), single women (IUI, n = 18,283; IVF, n = 5017), and same-sex couples (IUI, n = 4477; IVF, n = 816).

Information regarding the exposed and unexposed cohorts in the IUI and IVF treatment populations can be found in Table 1. Notably, single women were the oldest group in both treatment populations, with 38 (25th–75th percentile, 35–40) years for the IUI population and 40 (25th–75th percentile, 38–42) years for the IVF population. In general, the women in all the exposed cohorts were more overweight than those in the unexposed cohort in the IUI and IVF groups.

A total of 39.92% of male partners of women undergoing IUI treatments with partner sperm had normal sperm quality, while for the IVF treatments group, the percentage was 33.01%. Most of the male partners of women undergoing ART treatments using donor sperm had a diagnosis of azoospermia (Appendix A).

### 3.2. Live Births per IUI Treatment Cycle

In the IUI population group, the live birth percentage was 12.3% in women using partner sperm. The live birth percentages and respective confounder-adjusted ORs (aORs) in the exposed cohorts were 16.6% and aOR 1.48 (95% CI: 1.38–1.59) in women with male partners using donor sperm compared with women using partner sperm, 11.7% and aOR 1.20 (95% CI: 1.13–1.28) in single women compared with women using partner sperm, 17.6% and aOR 1.46 (95% CI: 1.32–1.62) in women with female partners using donor sperm compared with women using partner sperm (Table 2).

### 3.3. Live Births per IVF Treatment Cycle

In the IVF population group, the live birth percentage was 25.7% in women using partner sperm. The live birth percentages and respective aORs in the exposed cohorts were 26.3% and aOR 1.16 (95% CI: 1.02–1.32) in women with male partners using donor sperm compared with women using partner sperm, 17.2% and aOR 0.88 (95% CI: 0.80–0.96) in single women using donor sperm compared with women using partner sperm, and 28.7% and aOR 1.20 (95% CI: 1.00–1.44) in women with female partners using donor sperm compared with women using partner sperm (Table 2).

For the IUI population group, the sensitivity analysis, which included the infertility diagnosis as a covariate, confirmed the results observed in the main analysis. For the IVF population group, variations from the main analysis were present. Indeed, no difference was observed between single women and women using partner sperm, while for women with female partners, the chances were increased compared with women using partner sperm (Appendix A).

The sensitivity analyses evaluating the effect of male partner semen quality confirmed the main analysis results for both the IUI and IVF treatments, except for the single women group. Indeed, no difference was observed for single women undergoing IVF treatments compared with women using good-quality partner sperm (Appendix A).

### 3.4. Secondary Outcomes

The chances of biochemical and clinical pregnancy after each IUI or IVF treatment cycle followed the same pattern observed in the main analysis. The related data are presented in Appendix A.

## 4. Discussion

In this nationwide cohort study, which included data from 11 years of ART treatments, we found an association between the use of donor sperm during IUI treatments and an increased chance of a live birth for women with male partners using donor sperm and women with female partners. For single women, the effect was less pronounced. For IVF treatments, no association was observed for women with male partners using donor sperm and women with female partners when compared with women with male partners using partner sperm, while single women had a reduced chance of a live birth. The same patterns were also observed for the chances of biochemical and clinical pregnancies.

Most studies investigating fertility treatment outcomes using donor sperm only focused on IUI treatments. Moreover, they did not compare the results with treatments using partner sperm, which could be considered the standard patient group for which most ART procedures were developed [7,21,22,23,24].

Our study was one of the first studies that investigated differences between donor and partner sperm use during ART treatments, both IUI and IVF, with a particular interest in the outcomes for the different population groups using donor sperm. The few similar studies available observed results comparable to ours. A recent retrospective cohort study from 2021 compared the IUI outcomes of women with female partners with heterosexual women undergoing IUI using either partner or donor sperm. Their results paralleled ours, with an increased adjusted live birth odds ratio for women with female partners (aOR 1.59, 95% CI: 1.15–2.20) [25]. Another non-interventional, retrospective, multicenter cohort study from Spain [8] compared the live birth rate after IUI treatments with donor sperm in single women, women with female partners, and women with male partners against IUI treatments in women using partner sperm. In accordance with our results, they also observed a statistically significant increase in the live birth proportion using donor sperm for each group for the age group ≤37 years old (18.8% for women with male partners using donor sperm, 16.5% for single women, 17.6% for women with female partners, and 11% for women with male partners using partner sperm). From 38 years onward, no difference was observed. However, in this study, the pregnancy chance decrease observed in women using partner sperm could be overestimated and partially explained by the selection criteria. Indeed, for the group including women with male partners using partner sperm, they selected only couples in which the man had a sperm quality comparable to the donors. This means that this was the only group in which the reason for infertility was most probably of female origin. An Australian study from 2014 [26] compared the result of different infertility treatments in single women, women with female partners, and the general population attending their clinic. After artificial insemination treatments, they observed a statistically significant increase in the pregnancy rate for women with female partners compared with the general population, with 12.64 ± 8.93 and 8.04 ± 1.27, respectively, as the mean percentage pregnancy rates and standard deviation (SD). For single women, the percentage was lower but not significantly so (6.58 ± 5.02, mean percentage pregnancy rate and SD). After IVF treatments, they observed a significantly reduced pregnancy rate in the single women group compared with the general population (21.84 ± 1.67 and 29.55 ± 0.65, mean percentage pregnancy rates and SD). For women with female partners, the percentage was increased but not significantly so (34.04 ± 10.19, mean percentage pregnancy rate and SD). However, this study did not adjust for important factors such as age. They did not specify the composition of the general population either, which probably included women using donor and partner sperm. A UK study from 2015 [27] found a statistically significant live birth proportion increase in women with female partners compared with women registered for treatment with a male partner after IVF/ICSI treatments (38.9% and 28.8%, respectively, for women younger than 35; 28.7% and 23.3%, respectively, for all ages).

Our study showed that the use of donor sperm in IUI treatments was associated with improved pregnancy results. This was not surprising considering that even without a clear contraindication for IUI treatments, women with male partners using partner sperm probably have some fertility issues derived from both partners. Usually, IVF treatments are directly offered to couples with severe male infertility, leaving couples with potentially more severe female infertility problems in the IUI population. Indeed, women using partner sperm have lower pregnancy chances no matter the quality of the sperm used.

The positive association of using good-quality donor sperm observed in the IUI treatments was not present in the IVF treatments. The advancements in the treatment and collection of sperm for IVF and ICSI make the use of donor or partner sperm similar, leaving female infertility issues as the major player in pregnancy success. Women with male or female partners using donor sperm probably have infertility issues comparable to or even lower than women with male partners using partner sperm. The single women group, being the oldest and probably having more undiagnosed infertility issues, showed a reduced pregnancy chance. Indeed when adjusted for infertility diagnosis, the live birth rate was comparable to the reference group.

The hypothesized beneficial effect of using partner sperm could also increase the chances of women using partner sperm due to previous exposure to the paternal seminal fluid, even if the women are more sub-fertile than in the other groups. This could partially explain why we did not observe any difference after IVF treatments. Previous exposure to paternal seminal fluid could be more relevant during IVF treatments since the embryo is transferred to the uterus without direct exposure to any seminal fluid components. In contrast, during IUI treatments, the sperm is placed in the uterus, contributing to its direct stimulation, as well as to the stimulation of other reproductive organs [28]. Moreover, women using donor sperm do not have the chance of being repeatedly exposed to the same seminal fluid used during the ART treatments, reducing its positive effect.

However, even if present, this positive effect of seminal fluid does not seem to play a major role, and without properly adjusting for other confounders, such as an infertility diagnosis, it is impossible to assess it definitively.

This study had several strengths. It was based on a large nationwide study population obtained from Danish health registers. The IVF register comprises data from public and private clinics that are obliged to report treatment cycles and biochemical and clinical pregnancy results [16,17]. We obtained the live births outcome data from the Danish Medical Birth Register, where all newborn children are registered in Denmark, and it is characterized by high completeness and validity [18,29]. Moreover, we included and controlled for important confounders, such as women’s age at the beginning of treatment, BMI, alcohol and smoking consumption, number and type of treatment, and the calendar year of the ART treatment. Thanks to our study design, we were able to have a complete follow-up of the study cohort. In addition, we retrieved our outcomes independently from the exposure status, preventing possible bias or outcome misclassification.

The study also had some limitations. As in any observational study, we cannot eliminate the presence of unknown confounders and residual confounding. Moreover, we did not have access to several sociodemographic and clinical variables, which could have been relevant because this type of data is not reported in the registers. Since Denmark has a public healthcare system, as a minimum, everyone who fulfills the criteria for treatments, such as age and BMI, is offered the same free-of-charge standard treatments. Patients with infertility problems suggested to be criminals or to have psychiatric problems are evaluated by experts before being accepted for treatment. However, there is no extra evaluation process for patients receiving donated sperm. According to Danish guidelines and legislation, couples can start treatment after one year of unprotected intercourse, while single women or women with a female partner can start treatment immediately. For this reason, our findings can be generalized only to other countries with similar systems.

We also had to remove approximately one-fifth of the treatment cycles in the analysis because we did not have information regarding the type of sperm used or relationship status, or there were inconsistencies in the data. Since these treatments were at risk of misclassification, we excluded them. However, in our analysis, we were able to take into consideration many different confounders that are known to have clinical relevance [20,30,31].

The female infertility diagnosis reported in the IVF register varied considerably depending on the partner with whom the women attended the clinics and the need to use donor or partner sperm. Indeed, for most single women and women with female partners, the main infertility reason given by the clinics was the absence of a male partner without further specifications. This is an important data limitation since the female fertility status seems to influence the result of the analyses.

The IVF register information regarding male infertility diagnosis is also lacking, and a considerable number of men have a diagnosis of unexplained infertility without information about sperm quality. Moreover, no information is present in the register regarding the parameters used to define “normal sperm quality”.

## 5. Conclusions

In conclusion, it is reassuring that ART treatment results in these new population groups using donor sperm are comparable to or even better than the results obtained in the more studied group of women with male partners using partner sperm. Lifestyle factors, such as BMI, seem different between the groups. Thus, it is important to start considering women requiring donor sperm as multiple groups with different needs and fertility issues. Healthcare professionals should pay attention to the screening, diagnosis, and reporting of infertility problems for all women and men attending the clinics. Studies like this contribute to building up the knowledge necessary to improve our ability to guide these new patient groups. However, more extensive prospective studies are needed to optimize the present fertility care, focusing on the type of sperm used and underlying infertility issues.

## Figures and Tables

**Table 1 jcm-12-02571-t001:** Descriptive characteristics of the study cohorts for IUI and IVF treatments of women with male partners using donor sperm, single women using donor sperm, women with female partners using donor sperm, and women with male partners using partner sperm from 2007 to 2017.

IUI Treatments					
Characteristics	Exposed Cohorts			Unexposed Cohort	*p*-Value
	Treatments in women with male partners using donor sperm (N = 8780)	Treatments in single women using donor sperm (N = 18,283)	Treatments in women with female partners using donor sperm (N = 4477)	Treatments in women with male partners using partner sperm (N = 80,949)	
No. of women	2638	5196	1251	27,756	
Age at time of treatment Median (25th–75th percentile)	34 (31–37)	38 (35–40)	33 (29–36)	33 (29–37)	<0.001
No. of treatments, n (%)					<0.001
1	2335 (26.6)	4957 (27.1)	1110 (24,8)	27,502 (34.0)	
2	1813 (20.6)	3827 (20.9)	900 (20.1)	20,493 (25.3)	
3	1402 (16.0)	2939 (16.1)	736 (16.5)	14,694 (18.2)	
4	1024 (11.7)	2180 (11.9)	560 (12.5)	7664 (9.5)	
5	758 (8.6)	1594 (8.7)	395 (8.8)	4395 (5.4)	
6	526 (6.0)	1115 (6.1)	283 (6.3)	2621 (3.2)	
6+	922 (10.5)	1671 (9.2)	493 (11.0)	3580 (4.4)	
Body mass index, n (%)					<0.001
<18.5 kg/m^2^ (underweight)	147 (1.7)	210 (1.2)	96 (2.1)	1720 (2.1)	
18.5–25 kg/m^2^ (normal)	4167 (47.5)	8580 (46.9)	2097 (46.8)	41,942 (51.8)	
25–30 kg/m^2^ (pre-obesity)	1824 (20.8)	4128 (22.6)	943 (21.1)	12,742 (15.7)	
30–35 kg/m^2^ (obese I)	893 (10.2)	1685 (9.2)	337 (7.5)	4773 (5.9)	
≥35 kg/m^2^ (obese II–III)	173 (2.0)	361 (2.0)	66 (1.5)	971 (1.2)	
Missing	1576 (18.0)	3319 (18.2)	938 (21.0)	18,801 (23.2)	
Smoking at the time of treatment, n (%)					0.177
No	6576 (74.9)	12,362 (67.6)	3048 (68.1)	52,445 (64.8)	
Yes	613 (7.0)	1260 (6.9)	305 (6.8)	5393 (6.7)	
Missing	1591 (18.1)	4661 (25.5)	1124 (25.1)	23,111 (28.6)	
Alcohol consumer, n (%)					0.047
No	3988 (45.4)	7489 (41.0)	1760 (39.3)	31,691 (39.3)	
Yes	3117 (35.5)	6037 (33.0)	1549 (34.6)	25,647 (31.7)	
Missing	1675 (19.1)	4757 (26.0)	1168 (26.1)	23,611 (29.2)	
Calendar year of treatment, n (%)					<0.001
2007–2009	1096 (12.5)	420 (2.3)	111 (2.5)	5964 (7.4)	
2010–2013	4460 (50.8)	6791 (37.1)	1550 (34.6)	36,039 (44.5)	
2014–2017	3224 (36.7)	11,072 (60.6)	2816 (62.9)	38,946 (48.1)	
IVF Treatments					
Characteristic	Exposed Cohorts			Unexposed Cohort	
	Treatments in women with male partners using donor sperm (N = 1937)	Treatments in single women using donor sperm (N = 5017)	Treatments in women with female partners using donor sperm (N = 816)	Treatments in women with male partners using partner sperm (N = 74,425)	
No. of women	890	1914	341	27,402	
Age at time of treatment Median (25th–75th percentile)	36 (32–39)	40 (38–42)	35 (32–38)	34 (31–38)	<0.001
Type of treatment, n (%)					<0.001
Fresh	1353 (69.9)	3919 (78.1)	558 (68.4)	53,052 (71.3)	
Frozen	584 (30.2)	1098 (21.9)	258 (31.6)	21,373 (28.7)	
No. of treatments, n (%)					<0.001
1	574 (29.6)	1822 (36.3)	320 (39.2)	27,318 (36.7)	
2	402 (20.8)	1179 (23.5)	211 (25.9)	17,405 (23.4)	
3	290 (15.0)	760 (15.1)	119 (14.6)	11,113 (14.9)	
4	210 (10.8)	486 (9.7)	64 (7.9)	6888 (9.3)	
5	141 (7.3)	297 (5.9)	46 (5.6)	4414 (5.9)	
6	101 (5.2)	189 (3.8)	28 (3.4)	2780 (3,7)	
6+	219 (11.3)	284 (5.7)	28 (3.4)	4507 (6.1)	
Body mass index, n (%)					<0.001
<18.5 kg/m^2^ (underweight)	24 (1.24)	68 (1.4)	14 (1.7)	1646 (2.2)	
18.5–25 kg/m^2^ (normal)	845 (43.6)	2445 (48.7)	387 (47.4)	41,408 (55.6)	
25–30 kg/m^2^ (pre-obisity)	469 (24.2)	1075 (21.4)	187 (22.9)	14,788 (19.9)	
30–35 kg/m^2^ (obese I)	231 (11.9)	489 (9.8)	45 (5.5)	5163 (6.9)	
≥35 kg/m^2^ (obese II–III)	34 (1.8)	87 (1.7)	11 (1.4)	742 (1.0)	
Missing	334 (17.2)	853 (17.0)	172 (21.1)	10,678 (14.4)	
Smoking at the time of treatment, n (%)					0.007
No	1496 (77.2)	3810 (75.9)	582 (71.3)	59,044 (79.3)	
Yes	98 (5.1)	239 (4.8)	30 (3.7)	4431 (6.0)	
Missing	343 (17.7)	968 (19.3)	204 (25.0)	10,950 (14.7)	
Alcohol consumer, n (%)					<0.001
No	988 (51.0)	2056 (41.0)	329 (40.3)	34,490 (46.3)	
Yes	587 (30.3)	1957 (39.0)	275 (33.7)	28,241 (38.0)	
Missing	362 (18.7)	1004 (20.0)	212 (26.0)	11,694 (15.7)	
Calendar years of treatment, n (%)					<0.001
2007–2009	54 (2.8)	79 (1.57)	18 (2.2)	3476 (4.7)	
2010–2013	855 (44.1)	1354 (27.0)	239 (29.3)	32,638 (43.9)	
2014–2017	1208 (53.1)	3584 (71.4)	559 (68.5)	38,311 (51.5)	

All values refer to IUI or IVF treatment cycles, except the variable “No. of women”, which gives an overview of the actual number of women included in the study.

**Table 2 jcm-12-02571-t002:** Crude and adjusted odds ratios (ORs) for live births in women with male partners using donor sperm, single women using donor sperm, women with female partners using donor sperm, and women with male partners using partner sperm in IUI and IVF treatments.

IUI Treatments		
Exposed Cohorts		Live Births
Treatments in women with male partners using donor sperm (N = 8780)	Yes, n (%)	1461 (16.6)
	No, n (%)	7319 (83.4)
	Crude OR (95% CI)	1.43 (1.34–1.52)
	Adjusted OR ^a^ (95% CI)	1.48 (1.38–1.59)
Treatments in single women using donor sperm (N = 18,283)	Yes, n (%)	2131 (11.7)
	No, n (%)	16,152 (88.3)
	Crude OR (95% CI)	0.94 (0.90–0.99)
	Adjusted OR ^a^ (95% CI)	1.20 (1.13–1.28)
Treatments in women with female partners using donor sperm (N = 4477)	Yes, n (%)	787 (17.6)
	No, n (%)	3690 (82.4)
	Crude OR (95% CI)	1.52 (1.40–1.66)
	Adjusted OR ^a^ (95% CI)	1.46 (1.32–1.62)
Unexposed cohort		
Treatments in women with male partners using partner sperm (N = 80,949)	Yes, n (%)	9933 (12.3)
	No, n (%)	71,016 (87.7)
IVF Treatments		
Exposed Cohorts		Live Births
Treatments in women with male partners using donor sperm (N = 1937)	Yes, n (%)	510 (26.3)
	No, n (%)	1427 (73.7)
	Crude OR (95% CI)	1.03 (0.92–1.16)
	Adjusted OR ^b^ (95% CI)	1.16 (1.02–1.32)
Treatments in single women using donor sperm (N = 5017)	Yes, n (%)	862 (17.2)
	No, n (%)	4155 (82.8)
	Crude OR (95% CI)	0.60 (0.55–0.65)
	Adjusted OR ^b^ (95% CI)	0.88 (0.80–0.96)
Treatments in women with female partners using donor sperm (N = 816)	Yes, n (%)	234 (28.7)
	No, n (%)	582 (71.3)
	Crude OR (95% CI)	1.16 (1.00–1.36)
	Adjusted OR ^b^ (95% CI)	1.20 (1.00–1.44)
Unexposed cohort		
Treatments in women with male partners using partner sperm (N = 74,425)	Yes, n (%)	19,108 (25.7)
	No, n (%)	55,317 (74.3)

a: adjusted for age at treatment, calendar year of treatment, BMI, smoking at the time of treatment, alcohol consumed, and no. of treatments. b: adjusted for age at treatment, calendar year of treatment, BMI, smoking at the time of treatment, alcohol consumed, type of treatment, and no. of treatments.

## Data Availability

According to Danish law, we are not allowed to distribute or publish patient data and make them available to others. However, other researchers may apply for access to data by sending an application to the Research Service at the Danish Health Data Authority.

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
