# Peer review of "Assisted Reproductive Technology Results Using Donor or Partner Sperm: A Danish Nationwide Register-Based Cohort Study"

_jcm, 2023, doi:10.3390/jcm12072571_

Round 1

Reviewer 1 Report

 a really comprehensive study with important information, in particular for clinicians in reproductive medicine

Author Response

Thank you very much for your comment.

Reviewer 2 Report

 The Authors carried out a Nationwide cohort study based on a wide Danish health register from 2007 until the end of 2017.

The main idea is interesting and relevant literature is not that rich

Nevertheless there are several points that need to be clarified. 

There is a serious question in terms of what populations the authors compare. 

The Group of “women with male partners using partners sperm” should be better described. How was the semen of such subjects? moreover, the methodology part is missing. 

Tables should include also statistically differences of The results.

Pleasen modify medically assisted reproduction (MAR) into assisted reproductive technology

Author Response

First of all, thanks for your comments and suggestions.

We included more information regarding the semen of the group of "women with male partners using partners' sperm". In the IVF-register there is no detailed information regarding semen concentration or other semen parameters. However, there is a variable indicating the infertility diagnosis given to the man by the doctors. This variable includes also diagnosis of aspermia, azoospermia, oliogozoospermia, oligoteratozoospermia, retrograde ejaculation. We made a supplementary table (table S3) showing the distribution of male infertility diagnoses. For privacy reasons and risk of identification, due to the low number of cases, we presented together data on patients with diagnosis of infection with HIV, missing or previous infertility.

We modified the methodology section (page 4), including information regarding the statistical analyses to assess differences in the covariates between the exposed and unexposed cohorts. We also made two sensitivity analyses to evaluate if the quality of partner semen could have an impact on the results. In one analysis, we only considered male partners with "normal semen quality" as our unexposed cohort. In the second one, we only used male partners with "reduced semen quality." The sensitivity analyses evaluating the effect of male partner semen quality confirmed the main analysis results except for the single women group. Indeed, no difference was observed between single women undergoing IVF treatments and women using good quality partner sperm (table S4).

We included the p-value in the Descriptive characteristics table (Table 1).

We also modified the term medically assisted reproduction (MAR) into assisted reproductive technology (ART).

Reviewer 3 Report

This is a valuable retrospective study, based on data from a national registry, presenting analysis of ART outcome of specific groups of women using partner or donor sperm. The results mat shed light on the additive value of exposure to seminal plasma in conjuction with IUI or IVF, considering also other factors that may inluence the results. The study is well decribed, larga dataset, the inclusion and exclusion criteria specified and results presented clearly. The strenghth and weaknessess of the study are mentioned. The results of this study add to the current knowledge base relevant to the subject of using donor sperm and merits.

Author Response

We appreciate your comments, and thank you.

Reviewer 4 Report

THE AUTHORS HAVE EXTRACTED USABLE DATA FROM THE NATIONAL DATABASE!!! Only this must be an example for all European nations!! have shown with simple and easily understandable methods how single patients do not need only semen as superficially may seem. this article, to date, must be an example for all centers and for all nations. in a very elegant and refined way, the superficial care given to patients is highlighted, You do not try to solve or understand the problem of infertility but they start to deals with techniques and stimulation without considering the factors that affect the outcome of techniques such as BMI, ovulation or hormonal problems and lifestyle.
Thanks authors for the elegance in saying these right and obvious remarks!

Author Response

Thank you very much for your comments and support of our study.

Reviewer 5 Report

This is a very interesting and well presented paper - congratulations.

Author Response

Thank you for your comment.

Reviewer 6 Report

Dear Authors,

I congratulate for this interesting article which could be defined an "hot topic".

I recommend the following comments:

1) Define MAR in abstract

2) why the sexual infections were not considered in the covariates?

3) critical: a serious limitation for your study, in my opinion, is the lack of semen parameters. In this case, results are strongly affected. Although a study on great numbers, the variability of semen is of undeniable importance when tackling such issue.

Author Response

Thank you for your comments.

Based on Reviwer 2 comment, we have changed the term medically assisted reproduction (MAR) to assisted reproductive technology (ART) and defined that accordingly in the abstract.

Regarding sexual infections, in the IVF-register, information is very limited. Infections such as HIV and hepatitis B and C are possible infertility diagnoses the patient can receive from the doctor. So we considered it in our sensitivity analysis when we included the infertility diagnosis as a covariate. However, as explained in the manuscript, the infertility diagnosis variable is not a very good information source in the register. We now specify in the manuscript that infectious diseases are included in the infertility diagnosis.

In the IVF-register there is no detailed information regarding semen concentration or other semen parameters. However, there is a variable indicating the infertility diagnosis given to the man by the doctors. This variable includes also diagnosis of aspermia, azoospermia, oliogozoospermia, oligoteratozoospermia, retrograde ejaculation. We included a list of all the male infertility diagnoses in the supplementary table S3. For privacy reasons and risk of identification, due to the low number of cases, we presented together data on patients with diagnosis of infection with HIV, missing or previous infertility. We also made two sensitivity analyses to evaluate if the quality of partner semen could have an impact on the results. In one analysis, we only considered male partners with "normal semen quality" as our unexposed cohort. In the second one, we only used male partners with "reduced semen quality.". The sensitivity analyses confirmed the main analysis results except for the single women group. Indeed, no difference was observed between single women undergoing IVF treatments and women using good quality partner sperm (table S4).

Round 2

Reviewer 6 Report

Already stated